# Women's experience of episiotomy: a qualitative study from China

Siyuan He [iD],[1,2] Hong Jiang,[1,2,3] Xu Qian,[1,2,3] Paul Garner[4]

SH and HJ contributed equally.

SH and HJ are joint first authors.

[1]School of Public Health, Fudan University, Shanghai, China
[2]Global Health Institute, Fudan University, Shanghai, China
[3]National Health Commission Key Laboratory of Health Technology Assessment, Fudan University, Shanghai, China
[4]Centre for Evidence Synthesis in Global Health, Department of Clinical Sciences, Liverpool School of Tropical Medicine, Liverpool, UK

**Correspondence to**
Professor Xu Qian;
xqian@shmu.edu.cn

## ABSTRACT

**Objective** To describe women's experience of episiotomy in urban China.

**Design** This is a semistructured, indepth interview with women after episiotomy. We analysed transcriptions using thematic analysis in Chinese. Emerging themes were debated in English to finalise interpretation.

**Setting** Two community health centres and four hospitals in Shanghai, China.

**Participants** Purposive sampling of 30 postpartum women who had experienced episiotomy; 25 were primiparous and 4 had deliveries by forceps. We interviewed health providers to complement the data.

**Results** We identified four main themes: (1) women's views of the procedure vary considerably; (2) pain interferes with daily life for weeks; (3) long-term anxiety is a consequence for some, described as a 'psychological shadow'; and (4) societal norms assume women will not complain.

**Conclusion** Women receive little information in advance about episiotomy, yet the procedure has a wide range of physical and psychological consequences. This includes long-term anxiety about the damage done to them as women.

## Strengths and limitations of this study

► This study is one of a few qualitative studies to explore women's experience of episiotomy after childbirth.
► The study identified an effect of episiotomy described in Chinese as a 'psychological shadow', and that societal norms meant women felt they were expected to suffer alone and not complain.
► We interviewed women at different times after episiotomy and were not able to evaluate whether their perceptions changed over time.

## INTRODUCTION

Doctors introduced episiotomy as a surgical procedure in the 1950s to reduce the risk of severe perineal tear, shorten delivery and prevent damage to the pelvic floor.[1] However, the procedure can cause pain in the immediate postpartum period, the wound can become infected, and the scar can cause long-term dyspareunia. Indeed, the benefits of routine episiotomy have been contested.[2] This balance between benefits and harms has been evaluated in randomised controlled trials. These are summarised in a Cochrane review which shows that there is no evidence that routine episiotomy has the benefits originally assumed and that more restricted use results in fewer women experiencing severe perineal or vaginal trauma.[3]

International institutions and professional societies now recommend episiotomy only when there is a clear clinical indication.[4–7] Practices in most European countries[8] show rates have fallen. However, episiotomy rates in vaginal births are still high in some countries, for example, 53.2% in Chile[9] (from hospital records), 73% from a hospital in Lebanon[10] and 92% from a hospital in Cambodia.[11]

Pushback from consumers in the early 1990s may have contributed to the decline in routine episiotomy in the UK, but in general recommendations for episiotomy have been set up mainly from the health provider's medical standpoint, with little reference to the views or preferences of women. A Cochrane review in 2017 pointed out that trials inadequately considered women's preferences, views on the procedures or the outcomes that are important to them.[3]

More recently, the WHO recognised the need for a 'positive childbirth experience',[4] which corresponds to the new Global Strategy for Women's, Children's and Adolescents' Health (2016–2030). With an increase in emphasis on women-centred outcomes in clinical decision making, women's experience of episiotomy is highly relevant.

In China, episiotomy used to be a routine practice for vaginal delivery.[12 13] In the last decade, hospital data reported rates of 47.4%–84.7%,[14–17] and some multicentre studies reported hospital rates from 41.2% to 69.7%.[18 19] For China, where there were 17.23 million births in 2016,[20] there could be as many as 7.33 million episiotomies a year (given a vaginal birth rate of 61.0% and an episiotomy rate of 69.7% among vaginal births).[18] Although the Chinese national obstetric guideline has recommended



restrictive use of episiotomy since 2016, it has not been implemented.[21]

We found no data in the published English literature on the experience of episiotomy in women in China, and we therefore carried out this study. Through this qualitative study, we aimed to describe women's experience of episiotomy in urban China.

## METHODS

### Approach, setting and sampling strategy

We used standard qualitative methods with semistructured, indepth individual interviews. The details of the methods were reported according to the Standards for Reporting Qualitative Research checklist (see online supplementary file 1).[22] We conducted the study in Shanghai (population of 24.2 million in 2016). The city has had a policy of routine episiotomy from 1999, with rates around 35.8%–86.67% from 2011 to 2014, and the restrictive use of episiotomy was recommended in 2015.[23–25] Community health centres hold pregnancy registration and information to allow home visits in the postnatal period for clients in their catchment area, while clinical services for childbirth are provided by higher level hospitals, and episiotomy practices vary between hospitals. Compared with general hospitals, maternal and child health (MCH) hospitals are more likely to adopt restrictive episiotomy policy since their midwives are experienced and well trained, and women at these settings tend to be at low risk. We used the two community health centres where we had worked previously and thus the staff were familiar with us: one in Pudong District, east of Shanghai (1459 pregnant women registered in 2017), and one in Xuhui District, west of Shanghai (775 pregnant women registered in 2017).

We used purposive sampling strategy, seeking women over 18 years old who had undergone an episiotomy in their last birth. We recruited women from three different postpartum periods (within 2 weeks, no more than 6 months, and 6 months or more after childbirth); we also took account of the types of hospitals to ensure a mix of experiences. Women being invited for this study delivered in various types of hospitals, including municipal MCH hospital, tertiary general hospital, district MCH hospital and secondary general hospital. Experienced healthcare providers who had over 3 years of work experience in maternal health area were recruited to confirm women's symptoms and help to better understand women's views and reflections. Two or three healthcare providers from each type of hospitals were involved in this study and their characteristics are shown in online supplementary file 2. We stopped interviewing women when we appeared not to identify new information.[26–28] The primary researcher (SH) carried out the interviews, under the guidance of the supervisors (HJ and XQ). SH is a master's student who had received training in qualitative methods and had a 6-month work placement with MCH administration.

All participants were informed about the research purpose and content. Interviews were conducted after

### Box 1    Interview guide

- ► What is your experience after episiotomy, from the childbirth to postpartum period? (Probe: discomfort, pain, swelling)
- ► Did episiotomy impact on your daily life? How? (Probe: walking, sitting, breast feeding, baby care, sexual life, medication, mood)
- ► How did you deal with your suffering or problems? (Probe: medical services usage)
- ► Are there some long-lasting effects of episiotomy you have noticed? If yes, what are they?

written informed consent was obtained from each participant.

### Patient and public involvement

When designing the study, we invited a few women to give us feedback on the approach and the questions to ask. We collected women's comments on the public internet forum and interviewed four women about the research topic before we designed the interview guide. This preliminary work led to several revisions to the interview guide.

### Data collection

We approached women by accompanying health staff during postpartum home visits or when women brought their children for child health check-up in community health centres between September 2017 and March 2018. We used an interview guide based on the literature and our research group discussions (Box 1).[29–31] We also reviewed women's comments on the public internet forum to improve the design of the interview guide and piloted the interview guide with four postpartum women. The piloted data were also included in our analysis as these were consistent with the main sample. Interviews were conducted in private rooms in the community health centres, hospitals and interviewees' homes, and all women provided signed consents. Interviews were in Chinese and recorded with permission. Health providers were recommended by relevant administrators and were invited to this study. They were interviewed at a private room in their workplaces.

### Data analysis

Medical master's students transcribed the interviews, and one of the interviewers (SH) checked them for accuracy. We used NVivo V.8.0 (QSR) software for thematic analysis.[32 33] Two researchers (SH and YC) read all the transcripts and coded the data to identify the reoccurring topics, ideas or concepts independently. After discussing the differences of the coding, they organised the data into initial themes. Initial themes and quotes were translated into English and checked by XQ. All the coauthors then further reflected on these themes and developed overarching categories, discussing the themes in both Chinese and English. The health professionals' responses were grouped against the emergent themes from the women's interviews and included within corresponding themes.

During this process, on two occasions we found themes that we could not translate directly into English. Rather than being a problem, these were both informative and underlying themes. The team discussed the words carefully in Chinese and English to gain a common understanding of meaning and cultural context.

The research team included three bilingual speakers (SH, HJ, XQ) and one native English speaker (PG). All the themes, descriptions and corresponding quotes were checked by all the authors.

### Reflexivity

As a team, we discussed our prior beliefs and experiences in early discussions and during analysis to reflect on how these may influence our analysis. The research team included people who had performed, repaired and experienced episiotomy (HJ, XQ, PG). Evaluating episiotomy and the uncertainty around benefits and harms is a topic of interest to all the authors, and, as with many medical and obstetrical interventions, we as researchers remain 'healthy sceptics'. Three authors have completed the Cochrane review examining this topic (HJ, XQ, PG) and reported that consumer views on the procedure are important for medical policy. All had experience in collecting and analysing qualitative data; PG and XQ have worked together for over 20 years on projects about whether obstetric practice and research evidence are in alignment in China.

### RESULTS

We interviewed 30 postpartum women, with age ranging from 21 to 40 (mean age 30.1) years. Twenty-five women were primiparous; all had experienced episiotomy and four also received assisted delivery with forceps. Seven women were interviewed within 2 weeks, 9 at 2 months and 14 at more than 6 months after childbirth (table 1). Four main themes emerged: (1) women's views of the

| Table 1 | Characteristics of postpartum women |
| --- | --- |
| | **Women** |
| Age (years) | |
| Mean±SD | 30.1±3.8 |
| Range | 21–40 |
| Parity | |
| Primipara | 25 |
| Multipara | 5 |
| Mode of delivery | |
| Episiotomy | 26 |
| Episiotomy with forceps | 4 |
| Interview time | |
| Within 2 weeks after childbirth | 7 |
| No more than 6 months after childbirth | 9 |
| 6 months or more after childbirth | 14 |

procedure vary considerably; (2) pain interferes with daily life for weeks; (3) long-term anxiety is a consequence for some, described as a 'psychological shadow'; and (4) societal norms assume women will not complain. The complete illustrative quotes are shown in online supplementary file 3.

### Women's views of the procedure vary considerably

This theme describes women's various views of episiotomy, including their knowledge, feelings and attitudes. The theme also explains how women's views are influenced from childbirth to postpartum period.

#### 'What is episiotomy?'

In general, women had little knowledge about episiotomy before childbirth, indicating that they were not well informed. Inadequate knowledge made women fearful of childbirth, but some women were informed and could articulate the benefits and harms and seemed to understand the justification for episiotomy. Nearly one-fifth of women knew very little about the procedure before childbirth. Some learnt about episiotomy during their labour, sometimes from other patients nearby. Others did not know what had been done until after the episiotomy, and some had just heard the name of this obstetric intervention, but did not know what it was.

> The doctors didn't inform me about the procedure (episiotomy). After childbirth, the woman in the same delivery ward asked me 'did you get episiotomy' and I reply 'what's the episiotomy?' I didn't know it before and I finally realized what the anesthesia and suturing meant at that time. (#9, 33 years old, primipara, 4 days after childbirth)

> …I used to wonder what episiotomy is, and only came to know exactly what it is after childbirth…at that time [When I was cut] I know it – Oh, this is episiotomy! (#14, 28 years old, primipara, 1 week after childbirth)

> At that time, I thought, 'Oh my god! They will certainly cut my vulva. The vulva would be ugly and [its function would be] affected!' It sounds scary. (#28, 21 years old, primipara, 6 months after childbirth)

A few women seemed to be more informed, from a variety of sources that included online materials, discussion with other women and from doctors. These more informed women were able to express the concept of balancing benefits and harms in their conversations:

> I think it is necessary to do episiotomy when it can accelerate the progress of labor. But if the baby can be delivered smoothly, episiotomy should be avoided. After all, it is still a surgical procedure. (#11, 30 years old, primipara, 2 months after childbirth)

#### Contrasting attitudes towards the policy of episiotomy

Women had differing opinions about the policy of episiotomy. Women's personal recovery experience was, unsurprisingly, important in shaping their views: some



clearly supported routine episiotomy, while others criticised this as excessive. One woman accepted routine episiotomy was required, and another multipara who had an episiotomy with her first childbirth requested it for her second delivery. These women had few problems with their current procedures and appeared to accept the need for the procedure. However, those who had a miserable experience seemed more likely to complain the negative effects and question the need for an episiotomy. Two quotes represent these different views:

> The hospital takes episiotomy as a routine practice during normal vaginal birth. I think if episiotomy can relieve your suffering, routine episiotomy should be recommended. I felt that my perineum recovered soon after episiotomy. On the other hand, episiotomy won't cause any big problems, as long as you move carefully and clean yourself frequently. (#26, 28 years old, primipara, 2 weeks after childbirth)

> The doctor said that my uterine contractions were too weak, but I didn't feel that way. I just needed some time. I don't like the episiotomy at all. I searched episiotomy on the Internet and found its rate in China is excessively high. Many situations are not necessary. The doctors might be afraid of potential risks. (#8, 34 years old, primipara, 2 months after childbirth)

### Pain interferes with daily life for weeks

This theme describes women's pain after episiotomy and how it influences their postpartum daily life widely.

#### Pain from episiotomy varied

Women's pain and discomfort varied, in some severe, and in a few lasted for months. Women in pain for 2 weeks only described the pain as 'a little pain or discomfort', but a few women reported considerable pain for months after childbirth, with three reporting this as 'intolerable' for more than 1 month. These women with severe pain also reported problems with suturing, including tight stitches, irritation from the stitches or the wound gaping.

> I still feel pain of my perineal wound now and I can feel the difference between the two sides of perineum…the right side with the episiotomy lack skin elasticity… (#1, 35 years old, multipara, episiotomy with forceps, 6 months after childbirth)

> The wound hurt in the first few days. Five days after delivery, I started to feel better, but I can still feel the pulling or tugging pain at the incision…it was a bit tight. (#26, 28 years old, primipara, 2 weeks after childbirth)

> The wound split at the six day after birth, then I suffered a lot because it recovered slowly. The pain had continued for half a month and the stitches cannot be absorbed…Even now, I am still feeling painful when I am sitting (#8, 34 years old, 2 months after childbirth)

### Restricted postures and movements

Avoiding pain and fear that the episiotomy would split meant women avoided moving around. Women stated they were conscious of the wound and were avoiding pain, so they had to walk or move slowly, or avoid contact as the wound hurt when pressed. Some had to sit or lie on one side or stay in one position for a long time to avoid pain, and this made them tired and uncomfortable. Three women with problems with healing of episiotomy complained that they could not sit down for a minute because of the horrible pain, which greatly influenced their postpartum life, such as sleeping and eating.

> At that time (half a month after childbirth), I couldn't sit or squat [because of the horrible pain], and I had to move very slowly. (#8, 34 years old, 2 months after childbirth)

> The healing was not very good [of my perineum]…in the first few days, I was fed by my mother. I couldn't sit [because of pain], and I just lay down there. I ate on the bed in the first month. (#20, 30 years old, primipara, episiotomy with forceps delivery, 2 months after childbirth)

### Obvious difficulties in breast feeding and defecation

The pain caused by episiotomy impacted on women's daily life in various ways. Among these, breast feeding and defecation were mentioned a lot. Several reported that pain from episiotomy interfered with breast feeding. Usually, women liked to feed their baby while sitting, but if this was painful they struggled to feed. Some of them learnt to breast feed by lying down or using breast pump in a standing position. Other women sat in pain and found it a struggle, increasing the difficulty and fatigue of breast feeding. Pain often interfered with defecation, with increasing pain and the sensation of the wound about to split while defecating. Just sitting or squatting was already hard. This fear of pain or that the wound would split open led women to avoid defecation, worsening existing postpartum constipation.

> It was very tiring and painful to sit down…I felt my wound was also swollen, and I had to sit on one-side, lean my body to the side without episiotomy. I sat in this way for the breastfeeding within the whole first month…this made my back hurt and sometimes it was really awful. (#28, 21 years old, primipara, 6 months after childbirth)

> My wound hurt very much in the first week, and I couldn't peep or poop at all because I couldn't sit on the toilet (This posture pulls the wound). Every time using the toilet was like a torture to me. I think that most women who have received an episiotomy would probably have the same problem as me. (#1, 35 years old, multipara, episiotomy with forceps, 6 months after childbirth)

Health providers had different views on the women's experience. They considered that perineal pain from

episiotomy is usually tolerable and does not last long, unless there is something wrong, such as an infection or stitches that could not be absorbed, which is very rare in their view. The doctors did not mention the effects on breast feeding. On the other hand, community healthcare providers and midwives confirmed the difficulties of breast feeding. Some commented that some women had to breast feed in a painful sitting position because they did not know how to feed the baby in any other way, and that the sitting posture was the proper way for the baby to suck the mother's nipples. Postpartum constipation and pain with defecation were all recognised as problems by the health professionals.

> If the wound gets infected because of improperly sterilization during the procedure, it would be very troublesome. The healing will take one to three weeks. In this kind of case, women with episiotomy would be more tortured than those with C-section. (Obstetrician, 28 years of relevant work experience, district MCH hospital)

> Episiotomy does have impacts on daily activities, such as breastfeeding. Some women are unwilling to breastfeed while lying down, or they just don't know how to breastfeed while lying down. Sometimes, people would feel anxious because of the pain. The milk secretion could also be affected by the pain. (Midwife, 20 years of relevant work experience, secondary general hospital)

### Long-term anxiety is a consequence for some, described as a 'psychological shadow'

Several women used the word 'psychological shadow' cast by the long-term effects of episiotomy. The Chinese word implies a negative experience of suffering or torment that leads to dread or worries of the future—a bit like the experience of war or a tumultuous personal event. In this research, the word 'psychological shadow' illustrates how episiotomy affected women fundamentally and long term. This word contained at least two mechanisms: the fear caused by a terrible experience made women avoid this again, and the other is that the miserable experience damaged women's confidence around sexuality. 'Psychological shadow' would continue through postpartum sexual life and the next childbirth in some women.

#### Undesirable and affected sexual life

The severe pain from the perineal wound made women fear sex. A woman even asked her husband to await until 1 year after childbirth because she suffered severe pain from episiotomy for nearly 2 months and feared sexual life might take her back to the nightmare again.

> Because of the terrible perineal pain, I asked my husband to resume sexual life a year later. I didn't dare to do it, because I worried the wound would pain again. (#16, 32 years old, primipara, 2 years after childbirth)

Painful experience after episiotomy also meant women imagined the operation had somehow 'changed' their sexual life in the future through physical damage. One woman said pain with sex might have arisen from her anxiety—the psychological shadow—instead of real physical pain. Some responses around resumption of sex and the 'psychological shadow' included beliefs that their vagina was damaged and had become "loose" and might not ever recover. For these women, they were unwilling to have sexual life and described being permanently 'changed' that there had been damage done to their vagina.

> Psychologically, I feel that the vagina cannot recover to original state…you feel the vagina is looser than before. And your spouse also has some psychological barriers to postpartum sexual life. I feel that many mothers who undergo episiotomy will have the shadows of sexual life more or less. The psychological shadow might disappear over time, but I don't know yet. (#1, 35 years old, multipara, 6 months after childbirth)

#### Less confidence in subsequent vaginal deliveries

The 'psychological shadow' also impacted on how women viewed a possible subsequent pregnancy. Women showed less confidence in subsequent vaginal deliveries and expressed their doubts through these questions: whether the episiotomy wound would hinder the process of the next vaginal delivery; whether the wound would split again in the next vaginal delivery; or whether they would be subject to another episiotomy. In some cases, 'psychological shadow' from episiotomy influenced women's willingness to have another child and brought obvious anxiety during further pregnancy: at least one woman claimed clearly that next time she would ask for a caesarean section to avoid episiotomy. One woman said: "if I had a vaginal birth again, and an episiotomy again. I cannot imagine what will happen, my vagina would be totally 'useless' for sexual life." The interviews indicated a high degree of anxiety about the long-term physical consequences and reflect how this then itself causes further anxiety. Another multiparous woman also said that she was deeply troubled by the fear of 'undergoing episiotomy again' during pregnancy.

> I don't dare to deliver my second child through normal birth (vaginal delivery). The experience of recovering from the episiotomy was indeed miserable. It really scared me. Maybe not having a second child is better…or maybe I would choose C-section even though it has some negative effects…if I had a vaginal birth again, and an episiotomy again. I cannot imagine what will happen, my vagina would be totally 'useless' for sexual life. (#13, 39 years old, primipara, 2 years after childbirth)

> The doctor directly did the episiotomy at my first childbirth. So I gained some childbirth experience



and I was always afraid that I would suffer episiotomy again during this childbirth. There was a psychological shadow when I thought of the childbirth…I was worried about these problems such as deliver again, episiotomy again, miserable recovery of episiotomy. Finally, I still got episiotomy again! (#29, 30 years old, multipara, 6 months after childbirth)

The 'psychological shadow' is not just about psychological issues since women indeed reported some physical problems. They mentioned the uneven or rough skin of perineal wound and painful intercourse, which affected the enjoyment of sexual life. Several multiparous women who experienced episiotomy twice reported that they need longer time to recover from the repeated episiotomy. The health providers were also aware of physical abnormalities following episiotomy and psychological concerns about sexual life. For further pregnancy and childbirth, health providers conceded that a miserable experience of episiotomy can cause women fear and anxiety over the next delivery, but they expressed different reflections on women's concern: most dismissed concerns about subsequent deliveries; one midwife, however, thought the hard scar left from the last episiotomy is easy to tear again and slower to heal, if episiotomy is not done in advance.

What are the impacts of episiotomy on further childbirth? It is true that it may cast a psychological shadow on those women. If the episiotomy wound from first childbirth is infected or she has a severe tear, she won't dare to have another child, or she might choose C-section. (Obstetrician, 28 years of relevant work experience, district MCH hospital)

It doesn't matter much because interval between births is generally long. It takes at least one year, right? The skin would recover within a year. (Midwife, 25 years of relevant work experience, district MCH hospital)

Some people are scar physique (a kind of people who easily have enormous scar). This kind of scar is hard and protuberant so that we fear the wound would tear again during the second childbirth. What's worse, If the scar tears and is sewed up again, it can't heal very well. (Midwife, 25 years of work experience, secondary general hospital)

### Societal norms assume women will not complain
There are specific social norms that pain and suffering are a necessary part of childbirth and a trial in women's lifespan. However, these norms ignore some serious cases so that some women undergo unfair criticisms and people-centred services are insufficient in relevant clinical practice and nursing.

### Pain from childbirth is normal and endurable
There are some societal norms or established opinions about vaginal delivery in society: the endurable pain or other discomforts are regarded as a normal part of childbirth and the puerperium, and is every woman's 'fate',

as an interviewee said. Pain and discomfort are expected to gradually disappear without treatment. Under these societal norms, women felt the expectations that they should not complain much about the 'slight and temporary discomforts', but to be strong and endure the pain or discomfort by themselves. While women accepted this, it appeared that this expectation did not take into account the more substantive pain, discomfort and interference with daily life associated with episiotomy (see our first theme) and this is distressing for women.

I would endure the pain and not mention it. It didn't hurt that much. I could still bear with it…it's normal thing, also the fate of every woman. (#29, 30 years old, multipara, 6 months after childbirth)

Whenever I said I felt sore of the perineal wound, they would say, 'why you still feel painful after 4 months?' It sounds like I shouldn't be sore. Every time my husband said these words, I would response to him, 'you should get a cut and experience the healing process'. (#28, 21 years old, primipara, 6 months after childbirth)

### Too many complaints incur criticisms
Women that complain a lot will be criticised and talked about by their families and others. Several women mentioned their family members expected them to endure 'a non-severe discomfort'. Indeed, two of them were frightened that if they complained too much they would be judged as 'being low-tolerant' (jiao qi). This word is a pejorative personality trait, which means a person exaggerates something that is slightly uncomfortable. This word refers to people who have 'weak minds' and who are rather cowardly. When women expressed or complained the postpartum suffering too much, their families thought that they were at risk of this weak character trait of being 'low-tolerant'. When these types of judgement happened, women felt upset and unwilling to speak out, suffering alone. Surprisingly, another interviewed woman even regarded the tolerance of pain as the only choice and even boast of her strong character. Thus, the societal norms make some women 'suffer alone' and stop them from seeking help.

I wondered if all the women would have the perineal pain after the childbirth…they [family members], such as my sister in law said that I was a bit low-tolerant…they all had birth experience but they never heard that a puerpera unable to sit down after childbirth…I didn't see a doctor because my families said every woman would experience pain after childbirth, and the doctor also said my wound healed well…At that time, I felt it was so hard to be a woman. (#16, 32 years old, primipara, 2 years after childbirth)

I'm not very low-tolerant…some women are too spoiled to bear any pain and they always groan, which I thought it is meaningless. Nobody could replace your sufferings. It's normal thing, also the fate of

every woman. (#29, 30 years old, multipara, 6 months after childbirth)

## Health services might be influenced by the societal norms

Indeed, these societal norms about tolerating pain also manifest in the way healthcare was provided. People-centred services were inadequate in the procedure and nursing of episiotomy. Most women thought suturing is more painful than being cut, yet some doctors did not check whether women were effectively anaesthetised during the suturing. One woman complained of pain during suturing but was told to 'wait-it will be finished soon', and another was told to stay still. One woman reported the pain was so severe she did move when being sutured, and then blamed herself for the subsequent healing problems because she had moved. The expected tolerance of pain extended to pain relief: a woman asked for pain relief after childbirth but was refused by the doctors with the reason that 'the level of pain after vaginal birth can be tolerated'.

> The suturing process was more painful. I cannot keep unmoved because the anesthetic effects tailed off later. And the doctor kept telling me not to move, saying that he couldn't sew up well if I still move. But it was painful and he was sewing up for a long time because my wound was very big…I couldn't stay still, and I didn't know whether the stitches were done properly. I don't know if it related to my unabsorbed suturing knot, maybe it resulted from my own body condition (some immune factor). (#20, 30 years old, primipara, episiotomy with forceps delivery, 2 months after delivery)

> I felt painful so much! I thought I really needed some treatments to relieve the pain but the doctor thought I could endure this kind of pain…I really can't endure it since my wound is very large. I hadn't fallen asleep for several days after childbirth. The pain was so awful! (#20, 30 years old, primipara, episiotomy with forceps, 2 months after childbirth)

## DISCUSSION

There are few qualitative studies on episiotomy worldwide, and the ones we have identified do not differentiate between episiotomy alone and perineal trauma (including episiotomy and severe tear). Most qualitative studies focused on episiotomy only were usually conducted in hospital settings and concerned with the shorter term consequences of episiotomy.[34–37] By contrast, we identified more information about women's perspectives and personal reflections in the community settings. The description of 'psychological shadow' seems an apt way to describe both physical and psychological consequences and how these play out together, for example with dyspareunia, where anxiety may worsen the physical experience.

Limitations of our study included, first, that some women were interviewed more than 6 months after childbirth, which might introduce recall bias towards their experiences shortly after episiotomy. Second, we interviewed women at different timelines after childbirth instead of performing at the three specified timelines for all participants, so we cannot know the duration of some women's suffering and the change in their understanding of episiotomy.

When we set out, we anticipated the healthcare providers would validate women's perceptions, but they seemed unaware of the long-term consequences and tended to underestimate the degree of pain and restricted function that women reported.

Episiotomy results in extensive physical discomfort for some and life troubles to solve. In this study, we confirmed women did suffer perineal pain or discomfort, consistent with qualitative and quantitative studies.[31 38 39] In addition, women in this study complained more about the unabsorbed stitches or split of stitches. Many reviews also indicate there are a few women who need removal or resuture services due to factors such as materials or skills.[40 41] Women in this study also reported that episiotomy limited postpartum daily activities, including sitting, breast feeding, defecation and intercourse.

We have found few studies reporting that episiotomy interferes with breast feeding. Chou et al[42] mentioned that perineal pain can interfere with the initiation of breast feeding and Persico et al[43] found that the exclusive breastfeeding rate of women with episiotomy on the first day after delivery was lower than the women with intact perineum. These physical symptoms or morbidity can also cause psychological burden or anxiety. A study in Jordan reported there was an association between postpartum depression and 15 health problems of obstetric, gynaecological (ie, episiotomy pain, infection) and general health conditions (including fatigue and headache).[44] These physical problems might have cumulative effects, as a prospective study indicated high burden of breastfeeding problems alone or with comorbid physical problems was associated with poor maternal mood at 8 weeks, while the high burden of physical health problems was not significantly associated.[45]

The influence on mood may also relate to sexual life and further delivery, and the impact on sex has been reported elsewhere.[46 47] Our findings highlighted that some women with episiotomy feared or wanted to avoid another pregnancy because of the pain they experienced, or that they would choose caesarean section in the next childbirth. This is consistent with other qualitative studies about vaginal childbirth.[48 49] A study from Turkey also indicated fear about impending childbirth can increase the likelihood of requesting a caesarean section.[50]

Episiotomy was administered in this study with women not even knowing it was going to happen. This lack of informed consent appears widespread and has been reported in other studies. One study in Brazil mentioned half of interviewed women did not receive any information

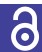

about the procedure before or during childbirth.[35] Another study reported that women were informed of the procedure but doctors lack the authorisation or even practised the procedure directly without any explanation.[36] Some women did not even know whether an episiotomy or spontaneous tear was done, and only noticed greater discomfort during suturing.[34] Women particularly lacked the knowledge about the consequences of episiotomy in our study, and one qualitative study about perineal trauma also identified the similar theme of 'being unaware of the episiotomy's consequences'.[35]

Women, their families and even some health professionals in this region also showed little understanding of some of the possible consequences of episiotomy. This opinion is consistent with a systematic mixed studies review about perineal trauma which reported on the theme 'normalization and feeling dismissed', which means women's health problems are regarded as a normal consequence of childbirth and that their questions were left unanswered by health professionals.[51] Some studies reported women felt frustrated and abandoned because they were 'dismissed by health care providers'.[52 53]

The study also raised the interplay between physical injury and pain, the societal expectations that this was normal, and the women's personal anxieties about the anticipated damage to their genitalia and anticipated pain with sex. When a woman with both physical pain and anxiety is not expected to complain, this can make matters worse. These factors and interactions are particularly important in China, where episiotomy rates remain high.

## CONCLUSION

Women were inadequately informed about episiotomy, but experienced consequences of the procedure, including pain and interference with daily life. These were compounded by social norms that expect them not to complain and the longer term anxiety about the physical and psychological effects on them as women.

**Acknowledgements** The authors thank the valuable contribution of the women and healthcare providers who participated in this study. The authors are very grateful to Qinjin Huang for her arrangements of study setting. The authors would like to thank Chunyi Gu and Yao Chen for the professional guidance and data analysis. The authors would also like to thank the following health staff for their assistance in recruitment: Xiaoxing Dai, Jiali Zhang, Hongxia Cui and Linlin Li.

**Contributors** SH, HJ and XQ designed the study and analysed data. SH and HJ drafted the paper, and XQ revised it. PG helped with analysis, commented on interpretation and helped write the manuscript. SH and HJ contributed equally to this study and should be regarded as co-first authors. All authors have verified and approved the final version of the abstract for publication.

**Funding** This work received support from the Effective Health Care Research Consortium, funded by UK aid from the UK government for the benefit of developing countries (grant: 5242).

**Competing interests** None declared.

**Patient and public involvement** Patients and/or the public were involved in the design, or conduct, or reporting, or dissemination plans of this research. Refer to the Methods section for further details.

**Patient consent for publication** Not required.

**Ethics approval** The research obtained approval from the Institutional Review Board of the School of Public Health, Fudan University (ID: 2017-12-0648).

**Provenance and peer review** Not commissioned; externally peer reviewed.

**Data availability statement** Data are available upon reasonable request.

**ORCID iD**
Siyuan He http://orcid.org/0000-0001-5375-1084

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
