## [Reviewer comments · BMJ Open]

ARTICLE DETAILS

TITLE (PROVISIONAL)	Women's experience of episiotomy: a qualitative study from China
AUTHORS	He, Siyuan; Jiang, Hong; Qian, Xu; Garner, Paul

VERSION 1 - REVIEW

REVIEWER	Lena Sagi-Dain Genetics Institute, Department of Obstetrics and Gynecology, Carmel Medical Center, affiliated to the Ruth and Bruce Rappaport Faculty of Medicine, Technion - Israel Institute of Technology, Haifa, Israel
REVIEW RETURNED	02-Sep-2019

GENERAL COMMENTS	As episiotomy is indeed a common procedure with unproven benefits and possible complications, efforts to explore any consequences of this procedure are highly needed. In addition, as the authors correctly state, evidence exploring the psychological effects of episiotomy is limited. Thus, the aim of the manuscript is relevant and important. The description of the "psychological shadow" and consideration of the social norms is novel and fascinating. However, the manuscript suffers from several crucial limitations. One of the most prominent restrictions of the study is the limited number of participants, not allowing to deduce any statistical calculations. Why only 30 women were chosen, and why did the authors focus on these specific participants? Is there any potential for selection bias? What do the authors mean in stating that "Individual interviews with postpartum women were stopped when data saturation was reached"? Despite the low number of participants, the authors should avoid using the words: "most", "some", "sometimes" in the results section, but rather state the exact numbers/percentages. Why did the authors choose the timelines for the interviews? Why weren't the interviews performed at the three specified timelines (i.e. two weeks, 6 months and 1 year) for all participants? Description of health providers' characteristics could also benefit the manuscript (age, gender, duration of professional experience) What were the "different episiotomy practices" in the four hospitals? The conclusions are far reaching considering the limited number of participants, possible selection bias and lack of presented rates. Finally, the manuscript could highly benefit from professional English language editor.
--

REVIEWER	Hannah Dahlen Western Sydney University
REVIEW RETURNED	20-Sep-2019

GENERAL COMMENTS	Women's experience of episiotomy: a qualitative study from China. Thank you for the opportunity to review this paper. The topic is of interest The strength and limitations of the study are not written well  • This study was one of the very few qualitative studies to understand details of women's experience, reflections and perspectives of episiotomy after childbirth Re write as • This study is one of a few qualitative studies to explore women's experiences, reflections and perspectives of episiotomy after childbirth These next three points either do not make sense or need editing and re-writing  • To enrich the information, we interviewed women of different times after episiotomy and included health care providers' responses as a supplement. • Some women were interviewed more than six months after childbirth, which might introduce the recall bias. • We interviewed the women in different times after childbirth instead of following up every woman from birth to recovery. Introduction This needs careful editing. For example you start sentences with And, there are many sentences that do not make sense and are grammatically incorrect Methods What theoretical framework was used? There are no references or description given of thematic analysis. Again this whole section needs a very good edit. You clearly do not understand reflexivity and this section needs to be re-written and referenced. Findings There are very few quotes in the paper and these should be woven throughout. Not dumped in a table. Discussion You make big assumptions for a qual study You need to organise the discussion better and it is riddled with grammatical errors
--

REVIEWER	Tahmina Begum icddr,b Dhaka, Bangladesh University of Queensland, Queensland, Australia
REVIEW RETURNED	26-Sep-2019

GENERAL COMMENTS	Thanks for giving me the chance to review this interesting paper. I add some comments, hope it will be helpful to improve the quality of
--

	articles 1. Introduction  • Justification is not clear enough, why women experience on episiotomy is essential to know particularly in China. • In the context of "secular trend" of Cesarean section in China, undue episiotomy might be a reason • what is the current practice of episiotomy ...% NVD does not indicate proportion of NVD as mode of birth as a whole • what is reported perineal tear rate? • Ultimately what message author wants to give; either promoting episiotomy with appropriate knowledge or restrictive episiotomy. the following reference might be helpful • Yamasato K, Kimata C, Huegel B, Durbin M, Ashton M, Burlingame JM: Restricted episiotomy use and maternal and neonatal injuries: a retrospective cohort study. Archives of Gynecology and Obstetrics 2016:1-6 • Baksu B, Davas I, Agar E, Akyol A, Varolan A: The effect of mode of delivery on postpartum sexual functioning in primiparous women. International Urogynecology Journal 2007, 18(4):401-406. • Serçekuş P, Okumuş H: Fears associated with childbirth among nulliparous women in Turkey. Midwifery 2009, 25(2):155-162. 2. Methods:  • Post-partum women were chosen purposively, I guess convenient sampling was adopted instead • If it is purposeful please correlate this with your study aim . • How the themes evolved , Inductive or deductive? • the guideline for data collection seems subjective & close ended! Researchers here biased about negative effect of episiotomy. I wonder either patient has chance to say having episiotomy relieve their prolonged labour at 2nd stage. What information they receive about episiotomy care and their usual clinical or non-clinical practice of episiotomy care are all seems important to know  • how the health care providers were selected need elaboration. what information they are validating ? 2. Result  • The number of health care providers if preselected should go under method section • The comparison of primi and multi parous patient missing • Is there any variation on women experience observed according to time gap since episiotomy being done? how the health care providers were selected need elaboration. what information they are validating .
--	--

VERSION 1 – AUTHOR RESPONSE

Response to reviewers' comments

Reviewer: 1 (Reviewer Name: Lena Sagi-Dain)

1. As episiotomy is indeed a common procedure with unproven benefits and possible complications, efforts to explore any consequences of this procedure are highly needed. In addition, as the authors correctly state, evidence exploring the psychological effects of episiotomy is limited. Thus, the aim of the manuscript is relevant and important. The description of the "psychological shadow" and consideration of the social norms is novel and fascinating. However, the manuscript suffers from several crucial limitations.

One of the most prominent restrictions of the study is the limited number of participants, not allowing to deduce any statistical calculations.

Thank you for the concerns and comments. We disagree. This is a qualitative study as stated in the title, and thus statistical analysis is an inappropriate method. The number of participants – the “information power” relates to obtaining sufficient to inform what we are investigating, sometimes called the “saturation point”. The methodology of sample size of qualitative study is discussed in many places, for example in this WHO publication: “When the method comprises interviewing of individuals, the minimum number should be higher (e.g. 10 or more) per study population, because the circumstances and behaviour of individuals are more variable”. (Collumbien M, Busza J, Cleland J, et al. Social science methods for research on sexual and reproductive health. Malta: World Health Organization 2012.)

Our study reached the final sample size according to the “saturation point” principle. Actually, many qualitative studies had similar sample size as our study, e.g. 21, 27, 7 and 11 participants in the following studies:

1. Laporte C, Vaure J, Bottet A, Eschaliere B, Raineau C, Pezet D, et al. French women's representations and experiences of the post-treatment management of breast cancer and their perception of the general practitioner's role in follow-up care: A qualitative study. HEALTH EXPECT. 2017;20(4):788-96.
2. Gillis C, Gill M, Marlett N, MacKean G, GermAnn K, Gilmour L, et al. Patients as partners in Enhanced Recovery After Surgery: A qualitative patient-led study. BMJ OPEN. 2017;7(e0170026).
3. Byrne V, Egan J, Mac Neela P, Sarma K. What about me? The loss of self through the experience of traumatic childbirth. MIDWIFERY. 2017;51:1-11.
4. Way S. A qualitative study exploring women's personal experiences of their perineum after childbirth: Expectations, reality and returning to normality. MIDWIFERY. 2012;28(5):E712-9.

2. Why only 30 women were chosen, and why did the authors focus on these specific participants? Is there any potential for selection bias?

Thank you for your comments. These women were selected according to purposive sampling strategy.

Purposive sampling is always used in studies using “qualitative” techniques (e.g. focus-group discussions or in-depth interviews), In such studies, it is usually preferable to purposively select individuals who will illuminate the research questions under study. There are some strategies for purposefully selecting “information-rich” cases in evaluation studies. One is “maximum variation sampling” whereby the researcher selects a small sample of great diversity. (Collumbien M, Busza J, Cleland J, et al. Social science methods for research on sexual and reproductive health. Malta: World Health Organization 2012.)

We adopted purposive sampling method, which highlight “information-rich”, to select the participants. We also chose “maximum variation sampling” strategy to attain a small sample of diversity. Therefore, the participants we recruited have various postpartum period, ages, parities and from different hospitals, it can provide rich information of experiences of episiotomy. We described it. (Page 7, 2nd Paragraph under the “Methods-Approach, setting and sampling strategy” section, Line 1-7)

3. What do the authors mean in stating that "Individual interviews with postpartum women were stopped when data saturation was reached"?

Thank you for your comments. It's a common technique to confirm that the number of participants has already satisfied the requirement of "information power" in qualitative study. Please refer to the reply of Question 1. The size of purposive samples typically relies on the concept of "data saturation." It is first defined as "no additional data are being found whereby the researcher can develop properties of the category. (Glaser BAAS. The discovery of grounded theory: Strategies for qualitative research. New York: Aldine Publishing Company 1967). A more general notion is "When new information produces little or no change to the codebook" (Guest G, Bunce A, Johnson L. How Many Interviews Are Enough? *Field Method* 2016;18(1):59-82.) or "qualitative studies should continue to collect data from groups or individuals until no new information of importance is encountered." (Collumbien M et al. reference above). To make it clearer, we have explained this point and cited some references as the following:

We stopped interviewing women when we appeared not to identify new information. (Page 8, 2nd Paragraph under the "Methods-Approach, setting and sampling strategy" section, Line 11-12)

4. Despite the low number of participants, the authors should avoid using the words: "most", "some", "sometimes" in the results section, but rather state the exact numbers/percentages.

Thank you for the suggestion. We think you are applying quantitative method to this data set and it is not appropriate. Based on our knowledge, the semi-quantified approach may help readers to have better understanding of the raw data. This approach is suggested in the paper focusing on the approach presenting qualitative research by Anderson (Anderson, 2010), in which the author encouraged that "the interpretation should be grounded in interviewees or respondents" contributions and may be semi-quantified, if it is possible or appropriate, for example, "Half of the respondents said ..." "The majority said ...". (Line 3-7, Page 5, Anderson C. Presenting and evaluating qualitative research. *Am J Pharm Educ.* 2010; 11; 74(8):141.).

We revised part of these words by replacing vague words such as "some" with semi-quantified words such as "more than one third of".

5. Why did the authors choose the timelines for the interviews? Why weren't the interviews performed at the three specified timelines (i.e. two weeks, 6 months and 1 year) for all participants?

Thank you for your suggestions. Based on the literature review, timeline was a factor associated with different health outcomes of episiotomy. Therefore, we designed to recruit women with different postpartum timelines to understand the experience of episiotomy, and we used standard qualitative approaches of purposive sampling to assure that we have achieved this. We reported this. (Page 7, 2nd Paragraph under the "Methods-Approach, setting and sampling strategy" section, Line 1-5)

In addition, women with various timelines can provide more information. For example, the topics about next pregnancy and long-term effects on sexual life were from women after childbirth more than one year.

6. Description of health providers' characteristics could also benefit the manuscript (age, gender, duration of professional experience)

Thank you for your suggestion. The characteristics of health care providers have been added in the Supplementary File 2.

7. What were the "different episiotomy practices" in the four hospitals?

Thank you for your comments. The both policies of routine episiotomy and restrict episiotomy are used in the research hospitals. Compared with general hospitals, MCH hospitals are more likely to adopt restrict episiotomy policy since there are more well-trained midwives and women at these settings tend to have less pregnancy complications. To make it clear, we have added this explanation in the manuscript. (Page 7, 1st Paragraph under the "Methods-Approach, setting and sampling strategy" section, Line 8-11)

8. The conclusions are far reaching considering the limited number of participants, possible selection bias and lack of presented rates.

Thank you for your comments. These issues are explained in Question 1-3. Because of the characteristics of qualitative studies, the conclusions always focus on some theories or interpretations rather than statistical presented rates.

Analysis of qualitative material is more explicitly interpretative, creative, and personal than survey analysis, since it is oriented towards explanation and discovery...By contrast to quantitative analysis, this approach does not begin by testing interrelations between different concepts or variables predetermined by a hypothesis – rather, it is inductive in style, which means drawing inferences from observations in the data to guide the construction of theory or hypothesis. It is not appropriate to conclude presented rates, as the interviewees are not statistically representative of the population but were sampled purposively (such as by age, marital status, distance to clinic). (Collumbien M, Busza J, Cleland J, et al. Social science methods for research on sexual and reproductive health. Malta: World Health Organization 2012.)

9. Finally, the manuscript could highly benefit from professional English language editor.

Thank you for your suggestion, you are correct. The manuscript has been rewritten by the English native co-author.

Reviewer: 2 (Reviewer Name: Hannah Dahlen)

1. The strength and limitations of the study are not written well

- This study was one of the very few qualitative studies to understand details of women's experience, reflections and perspectives of episiotomy after childbirth

Re write as

- This study is one of a few qualitative studies to explore women's experiences, reflections and perspectives of episiotomy after childbirth

These next three points either do not make sense or need editing and re-writing

- To enrich the information, we interviewed women of different times after episiotomy and included health care providers' responses as a supplement.
- Some women were interviewed more than six months after childbirth, which might introduce the recall bias.
- We interviewed the women in different times after childbirth instead of following up every woman from birth to recovery.

Thank you for your comments. We apologise for the poor writing: the submission needs more extensive rewriting by the English co-author. This has now been done. We have also rewritten these now.

- This study was one of a few qualitative studies to explore women's experience of episiotomy after childbirth
- The study identified an effect of episiotomy described in Chinese as a "psychological shadow", and that societal norms meant women felt they were expected to suffer alone and not complain
- We interviewed women at different times after episiotomy, and were not able to evaluate whether their perceptions changed over time

2. Introduction-This needs careful editing. For example you start sentences with And, there are many sentences that do not make sense and are grammatically incorrect

Thank you for your comments. We apologise for the poor writing: the submission needed more extensive rewriting by the English co-author.

3. Methods-What theoretical framework was used? There are no references or description given of thematic analysis. Again, this whole section needs a very good edit.

Thank you for your comments. Since there are few studies focused on women's experience of episiotomy, we designed this study as an exploratory study in order to understand women's views and experiences about episiotomy. In addition, the work came out of over 20 years work exploring aspects of obstetric care in China, and how it measures up against best practice defined in evidence-based systems.

We revised this section and cited some references of thematic analysis. (Page 9-10, 1st paragraph under the "Methods-Data analysis", Line 1-8)

4. Methods-You clearly do not understand reflexivity and this section needs to be re-written and referenced.

Thank you for your comments. We do understand reflexivity but perhaps not to the extent of you. We have now amended this section as:

As a team, we discussed our prior beliefs and experiences in early discussions and during analysis to reflect on how this may influence our analysis. The research team included people that had performed, repaired and experienced episiotomy (HJ, QX, PG). Evaluating episiotomy and the uncertainty around benefits and harms is a topic of interest to all the authors, and, as with many

medical and obstetrical interventions, we as researchers remain “healthy sceptics”. Three authors have completed the Cochrane review examining this topic (HJ, QX, PG) and reporting is that consumer views on the procedure are important for medical policy (Jiang H, Qian X, Carroli G, Garner P. Selective versus routine use of episiotomy for vaginal birth. *Cochrane Database Syst Re* 2017;8:2: CD000081). All had experience in collecting and analyzing qualitative data; PG and QX have worked together for over 20 years on projects about whether obstetric practice and research evidence are in alignment in China. (Page 11, “Methods-reflexivity” section, Line 1-10)

5. Findings-There are very few quotes in the paper and these should be woven throughout. Not dumped in a table.

Thank you for your comments.

It may be that you have a particular style of reporting qualitative research in mind, and we have another.

We would argue that our presentation follows best practice. Indeed the approach of using the table means that it is clear to the readers how the themes have emerged and the types of data from which they arose; the use of tables allow this to be more complete so we would argue this approach is more transparent as we substantiate our findings that adding in a few highly selected quotes into the text. It is also more efficient, and we are trying to ensure a short article that is easily read.

We checked this with our colleagues here including qualitative researchers who were authors of the Standards for Reporting Qualitative Research. The SRQR advises researcher to report linking evidence (e.g. quotes, field notes, text excerpts, photographs) to substantiate analytic findings (S17 in Table 1, O'Brien BC, Harris IB, Beckman TJ, et al. Standards for reporting qualitative research: a synthesis of recommendations. *Acad Med* 2014, 89:1245-51.). Another article also mentioned in the section of “synthesis of findings”, ‘Quotations from the articles may be included to illustrate the themes or constructs identified’ (Tong A, Flemming K, McInnes E, Oliver S, Craig J. (2012) Enhancing transparency in reporting the synthesis of qualitative research: ENTREQ. *BMC Med Res Methodol.* 12 (1):181).

To fulfil the requirements of including more quotes, we present the illustrated quotes in detail in a table, for the main text as a ready reference to illustrate quotes. However, we would request to keep the substantive table as an annex: the use of the table in this way not only makes clear how themes were developed but does so succinctly to align with the conventions of academic journals wishing to convey important substantiated messages within a specific word limit.

6. Discussion-You make big assumptions for a qual study. You need to organise the discussion better and it is riddled with grammatical errors.

We wrote the “Discussion” section according to the main findings and the reflections on the results by the team.

These data were stark: the pain, interference with daily life, as you can see from the quotes came through clearly; the idea of the psychological shadow is more subtle but important; and the issues over societal expectations to suffer in pain were also very clear from the data.

We have rewritten the discussion taking into account the comments.

Reviewer: 3 (Reviewer Name: Tahmina Begum)

Introduction

1. Introduction

- Justification is not clear enough, why women experience on episiotomy is essential to know particularly in China.

. In the context of "secular trend" of Cesarean section in China, undue episiotomy might be a reason

- what is the current practice of episiotomy ...% NVD does not indicate proportion of NVD as mode of birth as a whole

- what is reported perineal tear rate?

Thank you for your comments. Episiotomy had been a routine practice in China. We have had an estimate of the number of episiotomies conducted in China a year, which is 7.33 million. If some of these are unnecessary procedures, as the Cochrane review highlights, this would certainly be relevant to these women and to the whole of China. (Jiang H, Qian X, Carroli G, Garner P. Selective versus routine use of episiotomy for vaginal birth. *Cochrane Database Syst Re* 2017;8:2: CD000081)

As for "undue episiotomy might be a reason". We have not explained that well. We only found severe pain at delivery of the first baby may lead people to request CS at subsequent births. A mixed-methods systematic review indicated that "fear of pain associated with episiotomy" is one of the reasons for preference of cesarean section in China. (Long Q, Kingdon C, Yang F, et al. Prevalence of and reasons for women's, family members', and health professionals' preferences for cesarean section in China: A mixed-methods systematic review. *PLoS Med* 2018;15(10):e1002672.) However, we didn't find the definitive evidence that undue episiotomy is a reason for "secular trend" of Cesarean section.

In 2016 Chinese health national report, the rate of vaginal delivery is 61.0%, while the rate of Caesarean Section is 39.0%. We have provided this data in the "Introduction" section. (Page 6, 5th Paragraph under the "Introduction" section, Line 5)

As a latest systematic review reported: Perineal tear often occurs in vaginal childbirth, as reported 24% for second degree tears and 1.4% for obstetric anal sphincter injuries. (Aguiar M, Farley A, Hope L, et al. Birth-Related Perineal Trauma in Low- and Middle-Income Countries: A Systematic Review and Meta-analysis. *Matern Child Health J* 2019;23(8):1048-70.)

2. Ultimately what message author wants to give; either promoting episiotomy with appropriate knowledge or restrictive episiotomy. the following reference might be helpful

- Yamasato K, Kimata C, Huegel B, Durbin M, Ashton M, Burlingame JM: Restricted episiotomy use and maternal and neonatal injuries: a retrospective cohort study. Archives of Gynecology and Obstetrics 2016:1-6
- Baksu B, Davas I, Agar E, Akyol A, Varolan A: The effect of mode of delivery on postpartum sexual functioning in primiparous women. International Urogynecology Journal 2007, 18(4):401-406.
- Serçekuş P, Okumuş H: Fears associated with childbirth among nulliparous women in Turkey. Midwifery 2009, 25(2):155-162.

Thank you for your comments and references.

We do not wish to give an ultimate message about yes or no to routine episiotomy: this is a more complicated decision and should take into account the trial evidence contained in the Cochrane review. What we want to do is report women's perceptions or feelings about episiotomy. It is important in decision making about this procedure.

We carried out this study according to the research gap mentioned in the Cochrane Review of episiotomy [Jiang H, Qian X, Carroli G, Garner P. Selective versus routine use of episiotomy for vaginal birth. Cochrane Database Syst Re 2017;8:2: CD000081]. Moreover, the global strategies and recommendations are paying more attentions to women-centered outcomes, women's positive experience in childbirth and postpartum wellbeing.

Thank you very much for providing the relevant references. We have added the third reference to the discussion. Our research mentioned some women with terrible experience after episiotomy more likely to express the preference of selective C-section in next childbirth. It is similar to the study from Turkey that fear about impeding childbirth can increase the likelihood of requesting a caesarean section.

We revised this section (Page 5-6, 3rd and 4th paragraph under the "Introduction"), and cited one of the references you provided in the "Discussion" section.

3. Methods:

- Post-partum women were chosen purposively, I guess convenient sampling was adopted instead
- If it is purposeful please correlate this with your study aim.

Thank you for your comments. The postpartum women were chosen purposively. Based on the literature review, timeline was a factor associated with different health outcomes of episiotomy. Therefore, we designed to recruit women with different postpartum timelines to understand the experience of episiotomy. In addition, one strategy of purposive sampling is "maximum variation sampling" whereby the researcher selects a small sample of great diversity. (Collumbien M, Busza J, Cleland J, et al. Social science methods for research on sexual and reproductive health. Malta: World Health Organization 2012.) Thus, we took account of the types of hospitals, and we also recruited a certain percentage (at least one tenth) of multipara and women with forceps. We have modified this part in the manuscript. (Page 7, 2nd paragraph under the "Methods-Approach, setting and sampling strategy" section, Line 1-7)

4. Methods:

- How the themes evolved , Inductive or deductive?

Thank you for your comments. We organized the initial themes in an inductive way, and then we discussed and analysed the data deeply to get broader themes. To make it clear, we have made some revisions (Page 10, 1st paragraph under the “Methods-Data analysis” section, Line 2-5)

5. Method

- the guideline for data collection seems subjective & close ended! Researchers here biased about negative effect of episiotomy. I wonder either patient has chance to say having episiotomy relieve their prolonged labor at 2nd stage.

Thank you for your comments. The interview guideline we used in the study is open ended questions in Chinese and we did not notice this problem during translation. Now we have edited the English.

We would disagree that we are biased against episiotomy, although having done the Cochrane review. We probably wonder if routine episiotomy is the best policy: “healthy scepticism” maybe a better description. We did seek “positive” views about episiotomy and there were not many, although some women could understand that episiotomy might be of benefit. This is contained in the text. Our positionality is described in the reflexivity statement (Page 11, “Methods-reflexivity” section, Line 1-10).

We are not entirely clear that there is good evidence that routine episiotomy shortens the second stage, we would refer you to the Cochrane review.

6. What information they receive about episiotomy care and their usual clinical or non-clinical practice of episiotomy care are all seems important to know

This is a good question. As we concluded, women actually reported that they received little information as outlined in our first theme and this is part of the problem.

7. How the health care providers were selected need elaboration. what information they are validating ?

Thank you for your comments. We invited health care providers to the interviews to confirm or explain women’s symptoms and feelings of episiotomy. It’s triangulation method. We have elaborated the sampling process of health care providers. Please see our revisions as the following:

Experienced health care providers who had over three years work experience in maternal health area were recruited to confirm women’s symptoms and help to better understand women’s views and reflections. Two or three health care providers from each type of hospitals were involved in this study. (Page 7-8, 2nd paragraph under the “Methods-Approach, setting and sampling strategy”, Line7-11)

However, these interviews revealed little information and contribute little to the manuscript.

8. Result

- The number of health care providers if preselected should go under method section

Thank you for your comments. We have rearranged the relevant content after revisions. In our design, we planned to interview two or three health care providers from each type of hospitals. It has been written in the “Methods” section. The final number and characteristics of health care providers are detailed in a Supplementary file and the file is also cited in the “Methods” section now. (Page 7-8, 2nd paragraph under the “Methods-Approach, setting and sampling strategy”, Line 9-11)

9. The comparison of primi and multi parous patient missing

Thank you for your comments. Although we considered the parity as a factor that might be associated with the experience after episiotomy, however, we did not find a lot of difference between primiparous and multiparous women. Only one point was identified that some multiparous women with twice episiotomy mentioned they needed longer time to recover from the repeated episiotomy. We have added this information in the third theme (Page 18, 3rd paragraph under the “Results-Long term anxiety is a consequence for some, described as a “psychological shadow”, Line 10-11)

10. Result

- Is there any variation on women experience observed according to time gap since episiotomy being done?

Yes. There are variations on women’s experience observed according to time gap since episiotomy being done. For women being interviewed within two weeks after childbirth, pain and impacts on the daily life were mostly reported. For women six months or above since childbirth, sexual life and worry about the next delivery were the major concerns. We have described the experience after episiotomy with different time period across the ‘Result’ Section.

For example, when reporting the theme of “The pain interferes with daily life for weeks”, pain at different times was embodied as the following: “Women’s pain and discomfort varied-but in some was severe, and in a few lasted for months. Women in pain for two weeks only described the pain as “a little pain or discomfort”, but a few women reported considerable pain for months after childbirth, three reporting this as “intolerable” for more than one month.”

VERSION 2 – REVIEW

REVIEWER	Hannah Dahlen Western Sydney University Australia
REVIEW RETURNED	10-Feb-2020

GENERAL COMMENTS	There are still minor editorial issues which I presume will be picked up during processing I am still most concerned about the lack of quotes in the findings and just putting them all in the table is not usual practice in qual research. The researchers seem insistent on this.If this were done it would become apparent that some of these thematic headings don't
---

	fit the quotes. For example 'Women's knowledge of the procedure varies considerably' does not fit what is essentially women feeling they were not informed. It feels like more a content analysis than thematic analysis. Again under the societal norms theme there is more about pain and trauma than assuming women will not complain. Without the inclusion of some better quotes this just does not stack up.
--	---

VERSION 2 – AUTHOR RESPONSE

Response to reviewers' comments

I am still most concerned about the lack of quotes in the findings and just putting them all in the table is not usual practice in qual research. The researchers seem insistent on this. If this were done it would become apparent that some of these thematic headings don't fit the quotes. For example 'Women's knowledge of the procedure varies considerably' does not fit what is essentially women feeling they were not informed. It feels like more a content analysis than thematic analysis. Again under the societal norms theme there is more about pain and trauma than assuming women will not complain. Without the inclusion of some better quotes this just does not stack up.

Re: Thank you for your suggestions. We agree that that the findings can be better understood through revising interpretations and citing the suitable quotes under each theme. The representative quotes had already been woven from the table to the place under each theme throughout the results. After having re-reviewed the themes, we made some further adjustments and introduced some subthemes, especially the theme "Women's views of the procedure vary considerably" and "Societal norms assume women will not complain". Please see our revisions as following:

1. Women's views of the procedure vary considerably

This theme describes women's various views of episiotomy including their knowledge, feelings, and attitudes. The theme also explains how women's views are influenced from childbirth to postpartum period.

1.1 "What is episiotomy?"

In general, women had little knowledge about episiotomy before childbirth, indicating that they were not well informed. Inadequate knowledge made women under a kind of fear before childbirth, while sound knowledge capacitated women to realize both the benefits and harms so that they considered episiotomy more justly. Nearly one fifth of women knew very little about the procedure before childbirth. For some, their understanding of episiotomy was attained through their laboring experiences and the people around them. They even did not know what happened until other people told them, or just heard the name of this obstetric intervention.

"The doctors didn't inform me about the procedure (episiotomy). After childbirth, the woman in the same delivery ward asked me 'did you get episiotomy' and I reply 'what's the episiotomy?' I didn't know it before and I finally realized what the anesthesia and suturing meant at that time." (#9, 33 years old, primipara, four days after childbirth)

"... I used to wonder what episiotomy is, and only came to know exactly what it is after childbirth... at that time [When I was cut] I know it -- Oh, this is episiotomy!" (#14, 28 years old, primipara, one week after childbirth)

"At that time, I thought, 'Oh my god! They will certainly cut my vulva. The vulva would be ugly and [its function would be] affected!' It sounds scary." (#28, 21 years old, primipara, six months after childbirth)

A few women seemed to be more informed, from a variety of sources: online resources, discussion with other women, and from doctors. These more informed women were able to express the concept of balancing benefits and harms in their conversations:

"I think it is necessary to do episiotomy when it can accelerate the progress of labor. But if the baby can be delivered smoothly, episiotomy should be avoided. After all, it is still a surgery." (#11, 30 years old, primipara, two months after childbirth)

1.2 Two contrasting attitudes towards the policy of episiotomy

There were opposite opinions about the policy of episiotomy. Women's personal recovery experience was, unsurprisingly, significant in shaping their views: some clearly supported routine episiotomy, while others criticized this as an excessive obstetric intervention. One woman accepted routine episiotomy was required, and another multipara who had an episiotomy with her first childbirth requested it for her second delivery. These women had few problems with their current procedures appeared to accept the need for the procedure. However, those who had a miserable experience seemed more likely to complain the negative effects and question the need for an episiotomy. Two quotes below typically represent these two situations:

"The hospital takes episiotomy as a routine practice during normal vaginal birth. I think if episiotomy can relieve your suffering, routine episiotomy should be recommended. I felt that my perineum recovered soon after episiotomy. On the other hand, episiotomy won't cause any big problems, as long as you move carefully and clean yourself frequently." (#26, 28 years old, primipara, two weeks after childbirth)

"The doctor said that my uterine contractions were too weak, but I didn't feel that way. I just needed some time. I don't like the episiotomy at all. I searched episiotomy on the Internet and found its rate in China is excessively high. Many situations are not necessary. The doctors might be afraid of potential risks." (#8, 34 years old, primipara, two months after childbirth)

4. Societal norms assume women will not complain

There are specific social norms that pain and suffering is a necessary part of childbirth and a trial in women's lifespan. However, these norms ignore some serious cases so that some women undergo unfair criticisms and people-centered services are insufficient in relevant clinical practice and nursing.

4.1 Pain from childbirth is normal and endurable

There are some societal norms or established opinions about vaginal delivery in society: the endurable pain or other discomforts are regarded as a normal part of childbirth and the puerperium, which is every woman's "fate", as an interviewee said. This pain and discomfort are expected to gradually disappear without treatment. Under these societal norms, women felt the expectations that they should not complain much about the "slight and temporary discomforts" but to be strong and endure the pain or discomfort by themselves. Whilst women accepted this, it appeared that this expectation did not take into account the more substantive pain, discomfort and interference with daily life associated with episiotomy (see our first theme) and this is distressing for women.

"I would endure the pain and not mention it. It didn't hurt that much. I could still bear with it... it's normal thing, also the fate of every woman." (#29, 30 years old, multipara, six months after childbirth)

"Whenever I said I felt sore of the perineal wound, they would say, 'why you still feel painful after 4 months?' It sounds like I shouldn't be sore. Every time my husband said these words, I would response to him, 'you should get a cut and experience the healing process.'" (#28, 21 years old, primipara, six months after childbirth)

4.2 Too many complaints incur criticisms

Too many complaints are not expected and might incur criticisms or gossips. Several women mentioned their family members expected them to endure the "a non-severe discomfort". Indeed, two of them were frightened that if they complained too much they would be judged as "being low-tolerant" (Jiao qi). This word is a pejorative personality trait, which means a person exaggerate something that is slightly uncomfortable. This word refers to people who have "weak minds" and who are rather cowardly. When women expressed or complained the postpartum suffering too much, their families thought that the women were at risk of this weak character trait of being "low-tolerant". When these types of judgement happened, the women felt upset and unwilling to speak out, suffering alone. Surprisingly, another interviewed woman even regarded the tolerance of pain as the only choice and even boast of her strong character. Thus, the societal norms make some women "suffer alone" and stop them from seeking help.

"I wondered if all the women would have the perineal pain after the childbirth... they [family members], such as my sister in law said that I was a bit low-tolerant... they all had birth experience but they never heard that a puerpera unable to sit down after childbirth... I didn't see a doctor because my families said every woman would experience pain after childbirth, and the doctor also said my wound healed well... At that time, I felt it was so hard to be a woman." (#16, 32 years old, primipara, two years after childbirth)

"I'm not very low-tolerant... some women are too spoiled to bear any pain and they always groan, which I thought it is meaningless. Nobody could replace your sufferings. It's normal thing, also the fate of every woman." (#29, 30 years old, multipara, six months after childbirth)

4.3 Health services might be influenced by the societal norms

Indeed, these societal norms about tolerating pain also manifest in the way health care was provided. People-centered services were inadequate in the procedure and nursing of episiotomy. Most women thought suturing is more painful than being cut; yet some doctors did not check whether women were effectively anaesthetized during the suturing. One woman complained of pain during suturing but was told to "wait-it will be finished soon"; and another was told to stay still. One woman reported the pain was so severe she did move when being sutured, and then blamed herself for the subsequent healing problems because she had moved. The expected tolerance of pain extended to pain relief: a woman asked for pain relief after childbirth but was refused by the doctors with the reason "the level of pain after vaginal birth can be tolerated".

"The suturing process was more painful. I cannot keep unmoved because the anesthetic effects tailed off later. And the doctor kept telling me not to move, saying that he couldn't sew up well if I still move. But it was painful and he was sewing up for a long time because my wound was very big...I couldn't stay still, and I didn't know whether the stitches were done properly. I don't know if it related to my unabsorbed suturing knot, maybe it resulted from my own body condition (some immune factor). " (#20, 30 years old, primipara, EP with forceps delivery, two months after delivery)

"I felt painful so much! I thought I really needed some treatments to relieve the pain but the doctor thought I could endure this kind of pain... I really can't endure it since my wound is very large. I hadn't

fallen asleep for several days after childbirth. The pain was so awful!" (#20, 30 years old, primipara, EP with forceps, two months after childbirth)

In addition, we think our study used thematic analysis rather than content analysis after our group discussion and reviewing some references (Vaismoradi M, Turunen H, Bondas T. Content analysis and thematic analysis: Implications for conducting a qualitative descriptive study. *Nurs Health Sci.* 2013;15(3):398–405. doi:10.1111/nhs.12048). On the one hand, content analysis usually includes quantitative counts of the codes, while thematic analysis provides a purely qualitative, detailed, and nuanced account of data. On the other hand, the content analyst can choose between manifest and latent contents before proceeding to the next stage of data analysis, however, the thematic analysis researcher is mainly advised to consider both latent and manifest content in data analysis. We think we used the thematic analysis during our data analysis, by identifying both latent themes such as "The pain interferes with daily life for weeks;" and manifest theme such as "Societal norms assume women will not complain".